# DANCE: DIFFICULTY AND NOVELTY CO-DRIVEN EXPLORATION

## ABSTRACT

Reinforcement learning in environments with sparse rewards poses a significant challenge. Numerous exploration techniques strive to surmount this challenge by inciting agents to explore novel states. However, as familiarity with the environment burgeons, the novelty of states wanes, yielding an unguided exploration trajectory during later phases of learning. To surmount this quandary, this study posits that the difficulty of attaining a state functions as a more potent intrinsic motivational beacon, guiding the agent throughout the learning process. This difficulty signal encapsulates pivotal insights into the environment's underlying structure and the task's trajectory, a facet transcending the exclusive purview of state novelty. Subsequently, we introduce a reward prediction network to acquire a hybrid reward sourced from both state difficulty and novelty. Initially elevated for novel states, this reward progressively converges toward the state's inherent difficulty as visitations accumulate. This dynamic formulation assuages the scourge of catastrophic forgetting, shepherding the agent precisely across the learning odyssey. We establish the theoretical underpinnings of this reward mechanism as a distinct manifestation of reward shaping. It ensures the consistency between the learned policy and the original policy and additionally transforms the sparse reward problem into a dense reward problem, consequently accelerating the entire learning process. We evaluate the proposed Difficulty and Novelty Co-driven Exploration agent on several tasks with sparse rewards, and it consistently achieves satisfactory results.

## 1 INTRODUCTION

A fundamental challenge in reinforcement learning (RL) is how to trade-off between exploration and exploitation. An agent has to decide whether to greedily exploit what is already known to maximize the expected cumulative reward or explore the unknown environment to gather more information that helps find a potentially better policy. Although simple exploration strategies such as $\epsilon$-greedy action selection Mozer et al. (2018) and correlated Gaussian noise Lillicrap et al. (2015) work well on a wide range of tasks, they are inefficient in hard exploration tasks with sparse rewards such as the Atari game *Montezuma's Revenge*. Standard RL algorithms often fail to obtain even a single positive reward.

Recently, various exploration methods Bellemare et al. (2016); Ostrovski et al. (2017); Andrychowicz et al. (2017); Liu et al. (2019); Trott et al. (2019); Warde-Farley et al. (2019); Zhao et al. (2020); Badia et al. (2020); Hartikainen et al. (2020); Zhang et al. (2021a); Henaff et al. (2022); Mutti et al. (2022); Mu et al. (2022); Hou et al. (2025); Bagaria et al. (2025) try to address the sparse-reward problems. Among them, the novelty-based rewarding approaches in intrinsic motivation Ryan & Deci (2000) make remarkable progress by giving agents intrinsic rewards whenever they visit an unexplored or unexpected state, *i.e.*, novel states. Measuring the 'novelty' usually requires an additional model to evaluate the environmental states' statistical distribution, *e.g.*, the count-based exploration methods Bellemare et al. (2016); Ostrovski et al. (2017); Tang et al. (2017) count each visit to a state as a way to quantify its novelty.

One typical class of methods leverages an environment-related transition model's prediction error to measure state novelty. The prediction error is small if the agent encounters a familiar state. However, most methods of this type suffer from the "Noisy TV" problem Savinov et al. (2019)

in a stochastic or partially observable environment. Among works addressing this problem, the notable ones are episodic curiosity through reachability Savinov et al. (2019) and Random Network Distillation (RND) Burda et al. (2019). RND employs a fixed randomly initialized neural network's prediction error as an exploration bonus and achieves competitive performances on several hard exploration Atari games.

However, the intrinsic rewards mentioned above face the problem of gradually losing efficacy during the learning process. With continuous exploration, the agent becomes more and more familiar with the environment, and the novelty of each state gradually vanishes. In this process, the intrinsic reward slowly loses its guiding significance; thus, learning is only driven by extrinsic rewards, leading to the agent's undirected exploration in the later learning stage. The agent may forget valuable states after quickly getting familiar with the environment, which hinders itself from having a deeper understanding of the tasks in the environment, thus affecting the overall progress of task completion.

In this work, we propose a new intrinsic motivation mechanism to make up for the shortcomings of the existing novelty-based exploration methods while fully using its advantages in the early stage of learning. Besides state novelty, we utilize the difficulty of reaching a state as another evaluation criterion to guide the exploration. The difficulty level of a state is measured by the estimated difficulty signal an agent takes from the initial state to the target one. Unlike the state novelty, the difficulty levels of states will not vanish after repeated visits. More importantly, the state difficulty is continuously self-optimized during the learning process, automatically generating a learning curriculum according to the agent's current ability. In other words, the state difficulty acts as a stepping stone toward the states that are more difficult to reach, which is beneficial for designing an effective exploration strategy.

We further build a Reward Prediction Network (RPN) to provide intrinsic rewards for the agent. The network architecture takes inspiration from the RND network, including a fixed and randomly initialized target network and a predictor network. Unlike RND, the prediction error between the target network and the predictor network is fitted to the difficulty signal when reaching the current state instead of reducing to zero. The prediction error is high for novel states and decreases to the states' difficulty that the predictor has trained on. In this way, the novelty and difficulty of states are integrated into the prediction error, helping the agent not only find novel states but also better learn from the experience. The proposed intrinsic reward provides the agent with a lifelong exploration and exploitation guidance, which can work even without access to extrinsic rewards.

This work makes the following three main contributions:

- We design an intrinsic exploration mechanism DANCE considering both the difficulty and novelty of the states to learn exploration strategies that maintain efficient exploration throughout the learning process.
- We propose a reward prediction network to provide intrinsic rewards leveraging the state's difficulty and novelty. The intrinsic reward is high for novel states but decays with repeated visits, modulated by the state's difficulty signal.
- We theoretically demonstrate that the DANCE policy is equivalent to the policy that standard RL learns. We further illustrate that DANCE provides dense rewards and a curriculum to accelerate the learning process, explaining our mechanism's effectiveness in a principled way.

We evaluate DANCE in several hard-exploration environments. Extensive experimental analyses and comparisons demonstrate the effectiveness of the learning algorithm. Compared with the state-of-the-art methods, DANCE significantly improves the learning efficiency in the Maze environment with vector-based state space and achieves competing results in hard Atari games with image-based state spaces. To facilitate further studies on reinforcement learning in hard exploration tasks, the source code and all the experimental results of this work are attached in the supplementary.

## 2 RELATED WORK

The sparse reward phenomenon is common in many hard exploration tasks. Most previous works focus on improving the effectiveness of exploration. Simple heuristic methods such as $\epsilon$-greedy Volodymyr et al. (2015) or Gaussian control noise Mnih et al. (2016) work well on a wide

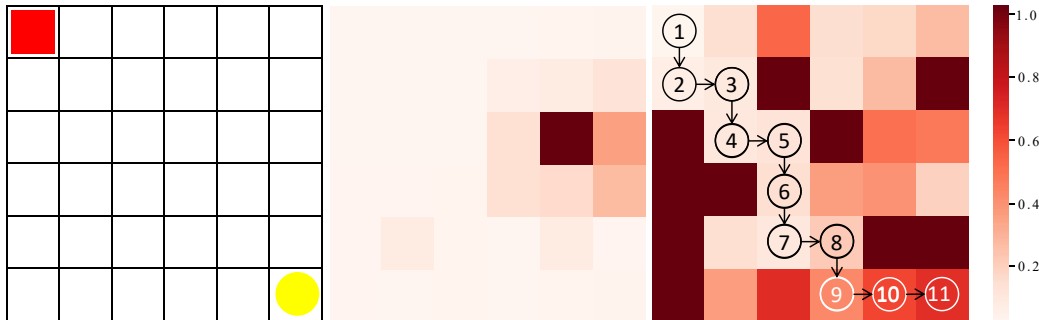

Figure 1: A simple maze environment to exemplify the differences between state difficulty and novelty: (Left) the environment, where the red square state is the initial state, the yellow circle one is the terminal state, and the agent can only move in the horizontal and vertical directions; (Middle) the heatmap of the novelty signal after training for 10K timesteps. and (Right) the difficulty signal's heatmap after training for 10K timesteps (the numbered arrows denote the final learned policy). The darker color indicates larger intrinsic rewards. Compared with the novelty signal, the difficulty signal greatly densities the rewards and automatically forms a path (the numbered arrows) to the target from easy to hard.

range of tasks, but they are inefficient in sparse reward environments such as those found in the Atari games like *Montezuma's Revenge* and *Venture*. In this section, we introduce some of the most related methods to our work on exploration in RL, including count-based exploration, curiosity-based exploration, and empowerment-based exploration.

**Count-based Exploration.** Count-based exploration Oudeyer et al. (2007); Strehl & Littman (2008); Silvia (2012) is a classical algorithm to solve hard exploration problems. These methods achieve good performance in tabular environments. To solve the hard exploration problems with image input, some works Bellemare et al. (2016); Ostrovski et al. (2017); Tang et al. (2017) adopt pseudo counts, density models, hash counts, and so on to measure the count for each visited state.

**Curiosity-based Exploration.** Recently, Curiosity-based exploration methods Pathak et al. (2017); Burda et al. (2019); Ghosh et al. (2019); Trott et al. (2019); Warde-Farley et al. (2019); Azizsoltani et al. (2019); Badia et al. (2020); Raileanu & Rocktäschel (2020); Zhang et al. (2021a;b); Henaff et al. (2022); Chen et al. (2022); Wang et al. (2023); Wan et al. (2023); Mahankali et al. (2024); Hou et al. (2025); Iten & Krause (2025); Bagaria et al. (2025) have yielded promising results for the hard exploration problems. Different kinds of curiosity models have been proposed, however, the basic idea behind these models is to give novel observations a bonus to encourage the agent to explore. Among them, RND Burda et al. (2019) is a simple-to-implement and easy-to-parallelize curiosity-driven exploration method. It employs the prediction error of a fixed randomly initialized neural network as an exploration bonus and outperforms average human performance in some games.

**Empowerment-based Exploration.** Empowerment-based exploration methods Mohamed & Rezende (2015); Liu et al. (2019); Leibfried et al. (2019); Kim et al. (2019) explore the environment by enhancing the agent's control over the environment. Go-Explore Ecoffet et al. (2021) combines the idea of depth-first search and empowerment and obtains amazing scores in the *Montezuma's Revenge*.

## 3   INTRINSIC MOTIVATION WITH DIFFICULTY

We introduce the difficulty of reaching a state as intrinsic rewards to guide efficient exploration even when the agent becomes familiar with the environment. The difficulty of a state adapts automatically as learning progress. As the policy converges, the state difficulty is stable no matter when or how many times the agent visits the state.

To illustrate the advantages of state difficulty over state novelty, we take a simple maze environment shown in Figure 1 (Left) as an example. In this environment, an agent at the initial state (red square in the upper left corner) tries to reach the terminal state (yellow circle in the lower right corner). We

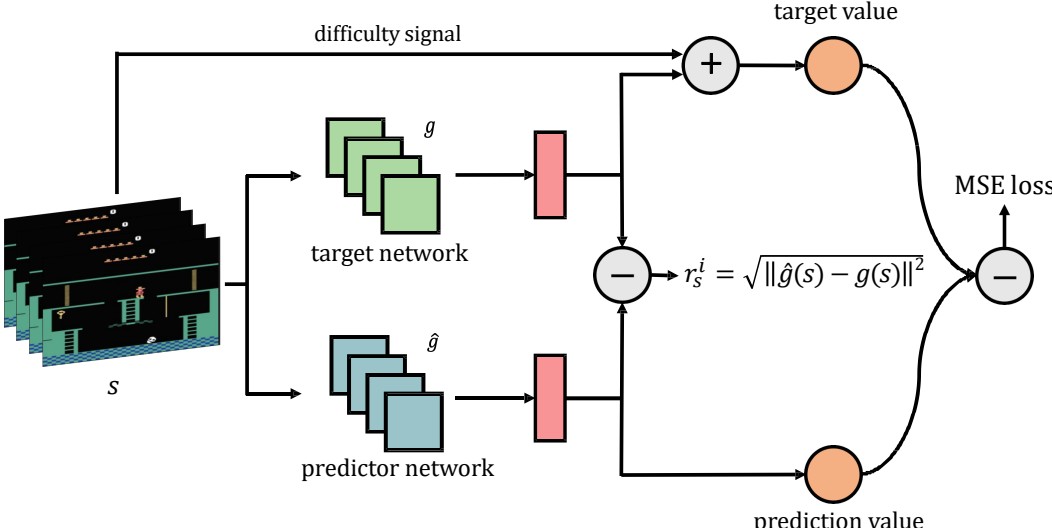

Figure 2: The RPN architecture consists of a target network $g$ and a predictor network $\hat{g}$. The current step number of state $s$ is normalized and then added to the target network's output to form the target value. The predictor network is trained on the data collected by the agent to predict the target value. The prediction error between the target and predictor networks is used as the intrinsic reward $r_s^i$.

dig into state novelty with heatmaps after training for 10K frames. As shown in Figure 1 (Right), the novelty of most states has become the same and decayed to zero. Unlike state novelty, the difficulty of each state converges to different values to densify the intrinsic rewards, revealing the environment structure and task direction. It helps the agent understand the environment and provides lifelong guidance to enable further exploration and a faster learning process. We further combine difficulty and novelty to drive the agent to explore the environment. Essentially, the novel or rewarded states can be seen as learning sub-goals, and the agent needs to transfer to a new state by choosing actions until reaching the sub-goal. The state difficulty can be seen as a type of depth-first search as Go-Explore Ecoffet et al. (2021). With the state difficulty, the agent learns faster to reach sub-goals. It forms environment structures and task directions to guide deeper exploration.

# 4 DANCE DRIVEN RL AGENT

The previous section introduces the state difficulty and proves its effectiveness in a simple motivating maze environment. The most common way to encourage exploration is to provide the agent with intrinsic rewards proportional to some quantitative measurements. In this section, we introduce DANCE intrinsic rewards which contains state difficulty and novelty.

Although difficulty signals can effectively help agents understand the environment, identifying such signals in complex environments poses an even more challenging problem. To solve this problem, we propose Reward Prediction Network (RPN) to model the difficulty and uncertainty of states. As shown in Figure 2, RPN consists of two neural networks: a fixed and randomly initialized target network $g : \mathcal{O} \rightarrow \mathcal{R}^k$, which takes observation to an embedding, and a predictor network $\hat{g} : \mathcal{O} \rightarrow \mathcal{R}^k$ trained on data collected by the agent. The difficulty signal of visiting a state $s$ is integrated into the target network's output as the target value. Thus, the prediction problem is set by the target value, and the predictor neural network $\hat{g}$ is then trained by gradient descent to minimize the expected mean-square error:

$$\mathcal{L} = \|\hat{g}\left(s\right) - \left(g\left(s\right) + ds(s)\right)\|^2, \tag{1}$$

where $ds(s)$ is the difficulty signal used for reaching this state. In the implementation, we initialize a vector with the same dimension as $g(s)$ and assign all elements in the vector the value of $ds(s)$.

The predictor network $\hat{g}$ learns to predict the sum of the output of the target network $g$ and the difficulty signal of the current state. We prove that the difference between output values of the well-

trained predictor network $\hat{g}$ and the target network $g$ on state $s$, *i.e.* $\|\hat{g}(s) - g(s)\|_2$, is equal to the normalized running average step number of state $s$. We thus define the intrinsic reward of reaching state $s$ as:

$$r_s^i = \|\hat{g}(s) - g(s)\|_2. \tag{2}$$

Our method derives the difficulty signal and enables the functionality of novelty-based methods in the early stages of the exploration. The prediction error is high for novel states while converging to the states' difficulty level after repeated visits. When the agent is unfamiliar with the environment, the prediction error for an unknown state is much higher than the normalized pseudo-step; thus, the agent is encouraged to explore the environment instead of staying still. In the later learning stage, the explored area's novelty has vanished, while the difficulty of state $ds(s)$ still guides the agent to enable effective exploration and efficient exploitation.

**Theorem 1.** *Suppose there are no two identical states in the state spaces, for a fixed policy $\pi$, i.e., using argmax to sample action, the intrinsic reward $r_s^i$ for state $s$ modeled by the RPN network is equal to the difficulty of state $ds(s)$.*

**Proof Sketch.** When the policy converges, the gradient of the Eq.( 1) loss function vanishes in the ideal case. The intrinsic reward $r_s^i$ aligns with the difficulty of state $ds(s)$. See Appendix for detail proof.

However, defining the difficulty of states constitutes a fundamental challenge. On the one hand, accurately specifying the difficulty level for each state is highly non-trivial and necessitates substantial expert knowledge. On the other hand, due to the non-zero property of state difficulty, incorporating such a non-zero reward function may lead to a deviation of the learning objective from the original reinforcement learning objective. To address this challenge, we propose that using proportional coefficient to the number of steps to reach a state as the difficulty signal is well-suited (e.g., $ds(s_t) \propto t$). When the policy converges, this simple yet strong signal exhibits the following properties: (1) For each state on the converged policy, the step count is fixed, and the intrinsic reward function is correspondingly fixed (align with the state's step number); (2) for states not on the converged policy, after sufficient exploration, their expected step count strictly exceeds that of states on the converged policy. The aforementioned properties effectively facilitate the agent's more efficient utilization of past experience and further encourages exploration of the environment. In the following section, we rigorously establish that the learning policy remains unbiased when the state difficulty is proportional to the number of steps required to reach the state under the converged policy.

## 5 THEORETICAL JUSTIFICATION

In this section, we first review a formulation of the intrinsic reward-based RL methods. Next, we demonstrate that the learning goal of the DANCE mechanism is the same as the original one with common RL algorithms, even though the intrinsic rewards in DANCE will not decrease to zero. Then, we argue that the dense rewards provided by DANCE help agents form a curriculum to accelerate learning. Finally, we provide the reward shaping form of DANCE, which further prove that the learned DANCE policy is consistent with the learned common RL policy.

### 5.1 FORMULATION OF INTRINSIC REWARD BASED RL

We consider the standard RL formalism that consists of an agent interacting with an environment to fulfill some tasks. An environment is described by a set of states $\mathcal{S} = \{s_0, s_1, s_2, \ldots, s_t, \ldots\}$, a set of actions $\mathcal{A} = \{a_0, a_1, a_2, \ldots, a_t, \ldots\}$, and a reward function $\mathcal{R}: \mathcal{S} \times \mathcal{A} \to \mathcal{R}$. After executing an action $a_t$ at a state $s_t$, the agent will transfer to a new state $s_{t+1}$ and get an extrinsic reward $r_t^e(s_t, a_t)$. For hard exploration tasks with sparse rewards, $r_t^e(s_t, a_t)$ is defined to represent the task goal only in a few states and set to zero for all other states. The goal of the RL agent is to learn a policy $\pi^*$ which maximizes the cumulative extrinsic rewards:

$$\pi^* = argmax_\pi \mathbf{E}_\pi \left[ \sum_{t=0}^\infty \gamma^t r_t^e(s_t, a_t) \right], \tag{3}$$

where $\gamma \in [0, 1]$ is the discount factor that trades off the importance of immediate and later rewards.

**Theorem 2.** *The cumulative extrinsic rewards received by policy $\pi^*$ should be no less than any policies $\pi^{any}$.*

*Proof.* Since the policy $\pi^*$ maximizes the cumulative extrinsic rewards, the cumulative extrinsic rewards obtained by any other policies $\pi^{any}$ should not exceed those of policy $\pi^*$. □

For the intrinsic reward-based methods, the agent will get two kinds of rewards: extrinsic reward $r_t^e(s_t, a_t)$ and intrinsic reward $r_{s_t}^i(s_t)$. The goal of the intrinsic reward-based agent is to learn a policy $\pi_i^*$ which maximizes the cumulative extrinsic and intrinsic rewards:

$$\pi_i^* = argmax_\pi \mathbf{E}_\pi \left\{ \sum_{t=0}^{\infty} \gamma^t \left[ r_t^e(s_t, a_t) + \beta r_{s_t}^i(s_t) \right] \right\}, \tag{4}$$

where $\beta > 0$ represents the weight of the intrinsic reward, and the intrinsic reward is far less than the extrinsic reward $r_{s_t}^i(s_t) << r_t^e(s_t, a_t)$.

**Theorem 3.** *The cumulative extrinsic rewards and intrinsic rewards received by policy $\pi_i^*$ should be no less than any policies $\pi^{any}$.*

*Proof.* Since the policy $\pi_i^*$ maximizes the cumulative extrinsic rewards, the cumulative extrinsic rewards obtained by any other policies $\pi^{any}$ should not exceed those of policy $\pi_i^*$. □

### 5.2 Connection between DANCE and Common RL

In this subsection, we further demonstrate that policy $\pi_i^*$ is at least a subset of policy $\pi^*$ when the intrinsic reward $r_{s_t}^i(s_t)$ is proportionate to the step number $t$ reaching state $s_t$ (e.g., $r_{s_t}^i(s_t) = \omega t$, and $\omega$ is a constant that satisfies $0 < \omega << 1$).

**Theorem 4.** *The policy $\pi_i^*$ DANCE agent learns is at least a subset of policy $\pi^*$ common RL method learns.*

**Proof Sketch.** According to Bellman Equation Sutton & Barto (1998); Sutton et al. (1999), for any RL problem, the policy $\pi^*$ is exist. See Appendix for detail proof.

Above all, we prove that policy $\pi_i^*$ is at least a subset of policy $\pi^*$. Here we assume that the MDP has an infinite horizon, since each finite MDP can be transformed into an equivalent infinite MDP, *i.e.*, an MDP which does not end at the terminal state but falls into a loop state that can never come out, so this proposition also applies to finite MDPs.

DANCE provides each state with a hybrid intrinsic reward mechanism including the difficulty and novelty signals using RPN. From RPN we intuitively find that this rewarded signal is non-zero for each visited state. It thus provides more dense rewards than methods with only the novelty signal. In addition, the reward signal does not disappear when the number of exploration times increases. These dense rewards help the agents guide the exploration more efficiently. Therefore, the DANCE mechanism provides the agent with more dense rewards to guide the exploration efficiently than standard novelty-based methods.

**Theorem 5.** *The DANCE mechanism helps to form a curriculum to accelerate learning.*

*Proof.* It is well known that humans learn better when training tasks are organized in a meaning-ful order, *e.g.*, by starting with easy ones and gradually progressing to more complex ones. The curriculum learning Bengio et al. (2009) paradigm inspired by this objective fact has shown good performances in supervised learning. In reinforcement learning, the intrinsic rewards of common novelty-based methods gradually vanish when the agent becomes more and more familiar with the environment, which can not provide more guidance for agents in the phase of learning policies to extrinsic rewards. DANCE helps the agent to explore the environment from easy to difficult. In the initial stage of exploration, DANCE gives intrinsic rewards according to the novelty of each state. During the phase of learning policies to extrinsic rewards, the intrinsic rewards given by DANCE for each state $s_t$ gradually fit the timestamps $t$. The intrinsic rewards of the states in a trajectory satisfy the following relationship:

$$r_{s_0}^i(s_0) \le r_{s_1}^i(s_1) \le \cdots \le r_{s_t}^i(s_t) \le \cdots . \tag{5}$$

According to the definition of curriculum learning Bengio et al. (2009), the state sequence $\{s_0, s_1, \ldots, s_t, \ldots\}$ of the DANCE mechanism forms a curriculum. □

Figure 1 (Right) provides an illustration of Theorem 5. The heatmap shows that the difficulty signals of the states are still distinct after the agent gets familiar with the environment, which demonstrates that DANCE indeed provides dense rewards and more guidance for better exploration and learning acceleration. Besides, the color of squares on the learned policy (denoted as the numbered arrows) are getting darker, indicating a series of curriculum from easy to difficult. Notably, Figure 1 (Middle and Right) validates our hypothesis: compared with the novelty signal, which gradually diminishes in the environment, the difficulty signal consistently maintains a high level for samples outside the reward trajectory, persistently facilitating the agent's exploration.

## 5.3 REWARD SHAPING FORMULATIONS

It is worth noting that most existing works Zhang et al. (2021b) suggest that intrinsic rewards need to satisfy *asymptotic consistency*, *i.e.*, intrinsic rewards should vanish after sufficient exploration. They argue that if the intrinsic rewards are asymptotically inconsistent, the final policy does not maximize the cumulative extrinsic reward, deviating from the goal of RL. Interestingly, our DANCE intrinsic reward is asymptotically inconsistent, but according to Proposition 4, the optimal policy the DANCE agent learns is equivalent to the optimal policy in the original MDP. This property can also be explained from the perspective of reward shaping.

The Policy Invariance Theorem Ng et al. (1999) states that a *potential-based reward shaping function* $F$ can guarantee the optimal policy learned by reward shaping will also be an optimal policy in the original MDP. Let $F$ be a potential-based reward shaping function if there exists a real-valued function $\phi : \mathcal{S} \to \mathcal{R}$, $F(s, a, s') = \gamma\phi(s') - \phi(s)$. Our DANCE intrinsic reward function is a potential-based reward shaping function, the potential function $\phi(s_t)$ is:

$$\phi(s_t) = \sum_{i=0}^{t} \frac{\omega i}{\gamma^{t+1-i}}, \qquad (6)$$

when $\gamma$ is 1, $\phi(s_t)$ is equal to $\frac{t(t+1)}{2}$. The DANCE intrinsic reward for state $s_t$ is:

$$r_{s_t}^i = \gamma\phi(s_t) - \phi(s_{t-1}) = \omega t. \qquad (7)$$

These analyses validate that DANCE not only accelerates agents' learning but also obtains the optimal policy in the original sparse reward problem.

## 6 EXPERIMENTS

To verify DANCE's effectiveness and generalization ability, we evaluate it on two distinct domains: 1) a simple maze; 2) seven hard exploration Atari games, including *Montezuma's Revenge*, *Pitfall!*, *Gravitar*, *Solaris*, *Venture* , *Frostbite*, and *Freeway*. We then present the performance results of various algorithms in the aforementioned environments devoid of extrinsic rewards to assess the pure exploration capability of the DANCE algorithm.

## 6.1 MAZE

To verify the validity of DANCE, we explore the pure exploration capability of DANCE. To verify this point, we compare different methods in a $50 \times 50$ Empty Maze environment. In this Maze environment, the maximum step number for one episode is set as 100. The implemented methods are the Deep Q Network (DQN) Volodymyr et al. (2015), DQN with Random Network Distillation (DQN+RND), DQN with NovelD Zhang et al. (2021b) (DQN+NovelD) and DQN with DANCE (DQN+DANCE). The hyper-parameter $\beta$ in RND, NovelD and DANCE is set to 0.01 and each algorithm is run for $100,000$ steps, and the explored area for each algorithm is depicted in Figure 3. We observe that the baseline DQN explores only a limited portion of the environment, covering only 314 different states (12.56% of the environment state coverage). The RND and NovelD algorithms perform similarly, each exploring 467, 470 different states (18.68% and 18.80% of the environment

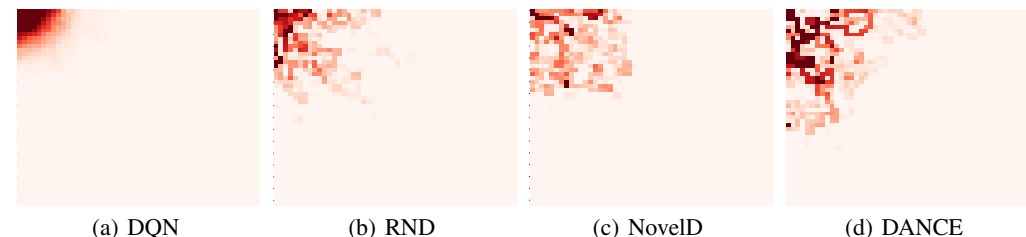

| (a) DQN | (b) RND | (c) NovelD | (d) DANCE |

Figure 3: The heatmap of DQN, RND, NovelD and DANCE on pure exploration maze in 100K steps. The initial state is located in the upper left corner.

state coverage), respectively. In contrast, DANCE, which incorporates a difficulty mechanism, significantly expands the exploration range compared to the other algorithms, exploring 617 different states (24.68% of the environment state coverage). These results confirm that DANCE's persistent difficulty signal effectively mitigates premature convergence, enabling systematic coverage of the state space. It is a critical advantage in pure exploration scenarios where sustained exploration drive is paramount.

## 6.2 HARD EXPLORATION ATARI GAMES

We convert the $210 \times 160$ input RGB frames to $84 \times 84$ gray-scale images for hard-exploration Atari games and feed the last 4 stacked gray-scale images into RPN. The hyper-parameter settings of DANCE are the same as those of RND Burda et al. (2019) ($\beta$ is set to 0.5), using the Proximal Policy Optimization (PPO) algorithm Schulman et al. (2017) with the RNN policy. The step number in DANCE is normalized by dividing the maximum step number in an episode set to $4,500$ in our experiment. Most experiments are run for 12.2K rollouts of length 128 per environment with 32 parallel environments, *i.e.*, for a total of 50 million steps (200 million frames) of experience.

To demonstrate DANCE's effectiveness, we compare its performance on Atari games with sparse rewards *Montezuma's Revenge*, *Pitfall!*, *Gravitar*, *Solaris*, *Venture*, and *Freeway* Bellemare et al. (2016) with RND and PPO. From Figure 6, DANCE learns much faster and better than RND in games *Montezuma's Revenge*, *Gravitar*, *Venture*, and *Frostbite*, while achieves a slight improvement in *Pitfall!*. It is a typical result in *Pitfall!* to fail to find any positive rewards, as the extrinsic positive reward is very sparse. DANCE outperforms RND and does not have the decline and instability as RND. These experimental results demonstrate that DANCE's intrinsic rewards encourage the agent to explore and exploit the environment more effectively than RND.

To further demonstrate DANCE's effectiveness, we then compare our DANCE with other five competitive exploration methods:AE-SimHash Tang et al. (2017), Exemplar Models Exploration (EX2) Fu et al. (2017), Curiosity-Driven Exploration (ICM) Pathak et al. (2017), Self-Imitation Learning Oh et al. (2018), and Exploration with Mutual Information (EMI) Kim et al. (2019) in seven Atari games. The final training performance of using 5 random seeds in 50M timesteps for

Table 1: Comparisons of representative algorithms with excellent performance on seven hard exploration Atari games. The results of AE-SimHash Tang et al. (2017), EX2 Fu et al. (2017), and EMI Kim et al. (2019) are from the original papers. All results are reported by using 5 random seeds in 50M timesteps.

|  | DANCE | PPO | AE-SimHash | EX2 | ICM | SIL | RND | EMI |
|---|---|---|---|---|---|---|---|---|
| Montezuma's Revenge | **5016** | 2461 | 75 | 0 | 1011 | 1259 | 3216 | 387 |
| Pitfall! | **−1.8** | −4.0 | − | − | − | − | −4.4 | − |
| Gravitar | **1631** | 1516 | 482 | 550 | 427 | 1629 | 1314 | 558 |
| Solaris | 1304 | 1054 | **4467** | 2276 | 2453 | 2854 | 1171 | 2688 |
| Venture | **1346** | 0 | 445 | 900 | 418 | 0 | 1150 | 646 |
| Frostbite | 3518 | 2067 | 5214 | 4901 | 4456 | 5873 | 2886 | **7002** |
| Freeway | **33.8** | 33.6 | 33.5 | 33.3 | 33.6 | 33.2 | 33.7 | **33.8** |

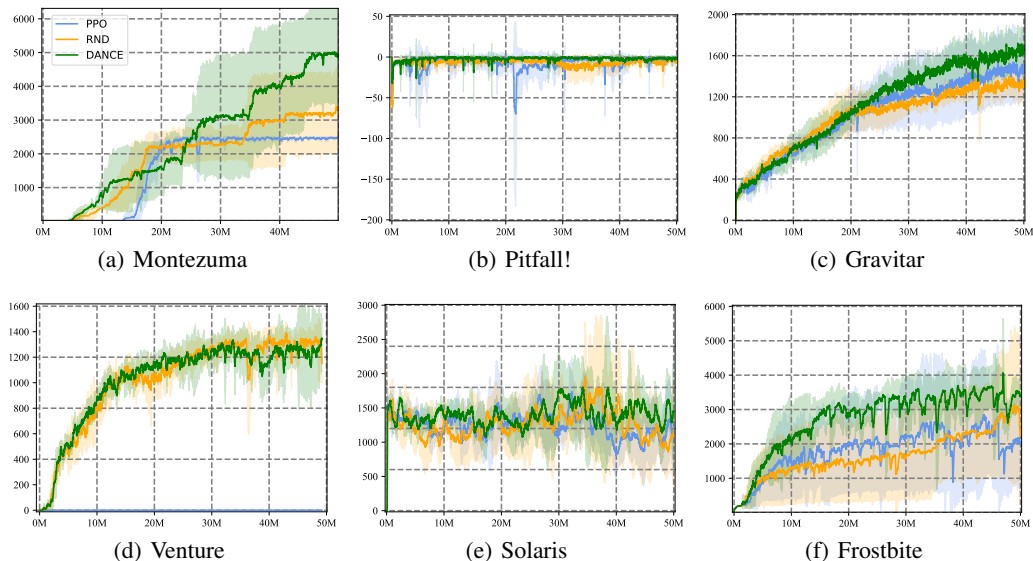

Figure 4: DANCE's performances on six hard exploration tasks compared to RND and PPO. The curves show the mean episodic return ($y$-axis) of five different seeds at each iteration ($x$-axis). The results are reported by using 5 random seeds in 50M timesteps.

each algorithm is listed in Table 1. From Table 1, we observe that DANCE achieves the best results in five of the seven environments. Taking *Montezuma's Revenge* as a canonical benchmark, we quantify the intrinsic rewards across the first and second rooms (0.047 for DANCE while 0.018 for RND), observing a statistically significant disparity in their mean values. DANCE fails to achieve competitive performance in *Solaris* and *Frostbite*. The highly similar visual backgrounds across these environments introduce significant interference to the intrinsic reward signal—specifically, the step number learning objective based on the Reward Prediction Network. This degradation stems from high background similarity between the environments, which substantially disrupts the intrinsic reward mechanism—specifically, the RPN-based step-wise learning process. Even so, DANCE made a significant improvement compared with other exploration methods. DANCE represents a significant advance in exploration methods, allowing the agents to navigate further and explore more diverse areas. This ultimately improves exploration efficiency, making DANCE a promising strategy for future research.

## 7 LIMITATION

Just as human beings easily get distracted by specific things and leave behind the task they are doing, we realize that the DANCE mechanism may face the same problem. For example, suppose the environment contains an observable clock. In that case, the predictor network may learn to look solely at the clock to predict the step number, thus may no longer yield any meaningful measure of progress influenced by the agent. This clock example poses a significant challenge for all the intrinsic reward based methods, and few of them can deal with this challenge well.

## 8 CONCLUSION

In this paper, we propose DANCE, an exploration mechanism which learns the difficulty signal of each state. By considering the difficulty and novelty of the state simultaneously, the intrinsic rewards provided by DANCE can continuously guide the agent to learn efficiently during the whole training process. Theoretical analysis shows DANCE constitutes a reward shaping framework for curriculum reinforcement learning. The detailed experimental results on the maze environment and some hard exploration games show that DANCE is beneficial for a wide range of RL algorithms. DANCE provides a simple yet strong baseline for solving hard exploration tasks.

## 9 REPRODUCIBILITY STATEMENT

To enhance the reproducibility of our work, we have made comprehensive efforts to document and share all relevant resources. The source code for our experiments is provided as supplementary material. Detailed experimental configurations, model hyperparameters, and procedural descriptions are thoroughly documented in the Appendix. Furthermore, complete proofs for theoretical claims and theorems presented in the paper are included in the Appendix to ensure verifiability.

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

# A APPENDIX

## A.1 ADDITIONAL EXPERIMENT

To further demonstrate the validity of DANCE, we evaluate DANCE and RND's performance in the absence of any extrinsic reward to see whether the proposed DANCE mechanism improves the exploration performance. We use *Montezuma's Revenge* as the experimental environment and set all the extrinsic rewards as 0. Figure 5 shows the mean episodic return and the mean number of rooms over the training process. From Figure 5 we see that the DANCE agent explores more rooms and receives a higher episodic return than the RND agent, which demonstrates the superiority of DANCE. In many practical applications, agents need to explore environments with extremely sparse rewards (or even no rewards), and the experimental results show that DANCE outperforms many competing algorithms. The furthest exploration of DANCE explores 6 rooms and recieves $3,600$ points, as shown in Figure 6. During the pure exploration training process, the DANCE agent successfully visit 11 rooms in total without any extrinsic rewards.

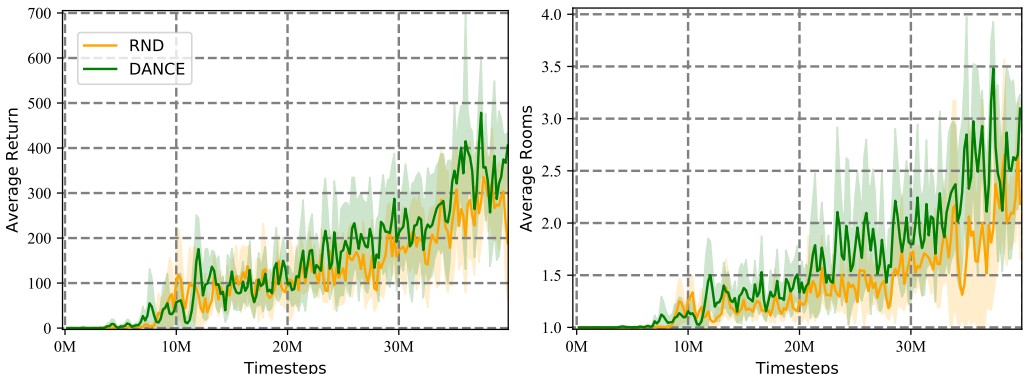

Figure 5: (Left) Mean episodic return and (Right) mean rooms visited received by the pure exploration agent in *Montezuma's Revenge* trained without access to extrinsic rewards using RND and DANCE. The results are reported by using 5 random seeds in 40M timesteps.

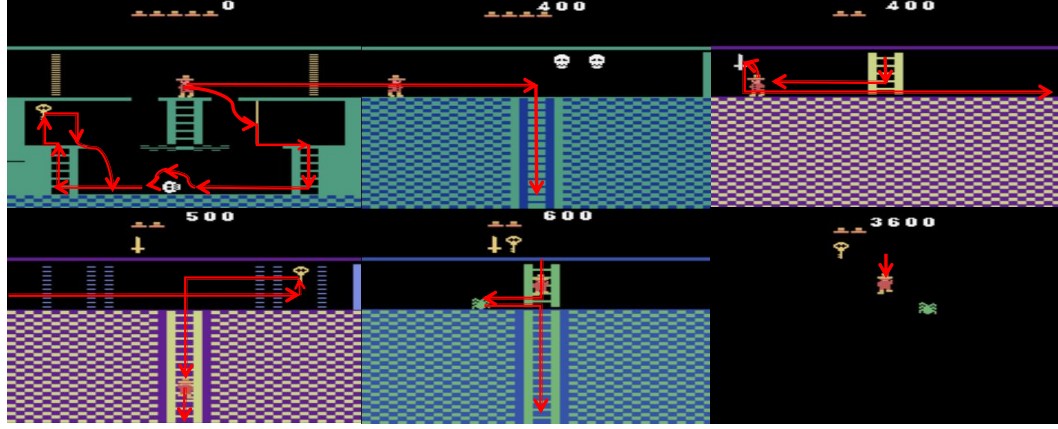

Figure 6: An example policy the DANCE agent learned. DANCE explores 6 rooms and recieves $3,600$ points without extrinsic rewards.

## A.2 ADDITIONAL PROOF

In this section, we extend the assumptions of Theorem 1 and Theorem 4.

**Theorem 1.** *Even when there are $n$ identical states $s$ in a fixed policy $\pi$, the timesteps for these states are $t_0, t_1, \ldots, t_{n-1}$. The difficulty signal $ds(s)$ of the state is the mean timestep of these states*

$ds(s) = \frac{t_0 + t_1 + \cdots + t_{n-1}}{n}$. *Under $n$ accesses, the sum of intrinsic rewards for $n$ identical states $s$ remains $t_0 + t_1 + \cdots + t_{n-1}$.*

*Proof.* The loss function $\mathcal{L}_s$ for state $s$ is:

$$\mathcal{L}_s = \sum_{i=0}^{n-1} \|\hat{g}(s) - (g(s) + t_i)\|^2 \tag{8}$$

$$= n(\hat{g}(s) - (g(s) + \frac{t_0 + t_1 + \cdots + t_{n-1}}{n}))^2 + C,$$

where $C$ is a constant. When the RPN converge, the optimal predictor network $\hat{g}(s)$ should minimize the loss $\mathcal{L}_s$ for the state $s$ and satisfy:

$$\hat{g}(s) - (g(s) + \frac{t_0 + t_1 + \cdots + t_{n-1}}{n}) = 0. \tag{9}$$

The intrinsic reward of state $s$ is:

$$r_s^i = \|\hat{g}(s) - g(s)\|_2 = \frac{t_0 + t_1 + \cdots + t_{n-1}}{n}. \tag{10}$$

Under $n$ accesses, the sum of intrinsic rewards for $n$ identical states $s$ remain $t_0 + t_1 + \cdots + t_{n-1}$.

$$n \cdot r_s^i = t_0 + t_1 + \cdots + t_{n-1}. \tag{11}$$

$\square$

It implies that for a fixed policy $\pi^{fix}$, the cumulative intrinsic rewards are the sum of the steps (e.g., $\mathbf{E}_{\pi^{fix}}[\sum_{t=0}^{\infty} \gamma^t \beta \omega r_{s_t}^i(s_t)] = \sum_{t=0}^{\infty} \gamma^t \beta \omega t$, when $\gamma = 1$. This ensures that Theorem 4 in the original paper remains valid even when multiple identical states in the fixed policy.

**Theorem 4.** *The policy $\pi_i^*$ DANCE agent learns is at least a subset of policy $\pi^*$ common RL method learns.*

*Proof.* According to Theorem 2, the cumulative extrinsic rewards received by optimal policy $\pi^*$ should be no less than $\pi_i^*$, we have :

$$\mathbf{E}_{\pi^*}\left[\sum_{t=0}^{\infty} \gamma^t r_t^e(s_t, a_t)\right] \geq \mathbf{E}_{\pi_i^*}\left[\sum_{t=0}^{\infty} \gamma^t r_t^e(s_t, a_t)\right], \tag{12}$$

and thus:

$$\mathbf{E}_{\pi^*}\left[\sum_{t=0}^{\infty} \gamma^t r_t^e(s_t, a_t)\right] + \sum_{t=0}^{\infty} \gamma^t \beta \omega t \tag{13}$$

$$\geq \mathbf{E}_{\pi_i^*}\left[\sum_{t=0}^{\infty} \gamma^t r_t^e(s_t, a_t)\right] + \sum_{t=0}^{\infty} \gamma^t \beta \omega t.$$

According to Theorem 1, for fixed policies $\pi_i^*$, the cumulative intrinsic rewards $\mathbf{E}_{\pi_i^*}[\sum_{t=0}^{\infty} \gamma^t \beta \omega r_{s_t}^i(s_t)]$ is equal to $\sum_{t=0}^{\infty} \gamma^t \beta \omega t$, hence we obtain:

$$\mathbf{E}_{\pi_i^*}\left[\sum_{t=0}^{\infty} \gamma^t r_t^e(s_t, a_t)\right] + \sum_{t=0}^{\infty} \gamma^t \beta \omega t \tag{14}$$

$$= \mathbf{E}_{\pi_i^*}\left\{\sum_{t=0}^{\infty} \gamma^t \left[r_t^e(s_t, a_t) + \beta r_{s_t}^i(s_t)\right]\right\}.$$

From Eqns. (13) and (14), we have:

$$\mathbf{E}_{\pi^*}\left[\sum_{t=0}^{\infty} \gamma^t r_t^e(s_t, a_t)\right] + \sum_{t=0}^{\infty} \gamma^t \beta \omega t \tag{15}$$

$$\geq \mathbf{E}_{\pi_i^*}\left\{\sum_{t=0}^{\infty} \gamma^t \left[r_t^e(s_t, a_t) + \beta r_{s_t}^i(s_t)\right]\right\}.$$

In addition, according to Theorem 3, the expected cumulative total reward $\pi_i^*$ gets should be no less than that of $\pi^*$. Thus policy $\pi_i^*$ also satisfies:

$$
\begin{aligned}
&\mathbf{E}_{\pi_i^*}\left\{\sum_{t=0}^{\infty}\gamma^t\left[r_t^e(s_t,a_t)+\beta r_{s_t}^i(s_t)\right]\right\} \\
\geq &\mathbf{E}_{\pi^*}\left\{\sum_{t=0}^{\infty}\gamma^t\left[r_t^e(s_t,a_t)+\beta r_{s_t}^i(s_t)\right]\right\},
\end{aligned}
\tag{16}
$$

and thus:

$$
\begin{aligned}
&\mathbf{E}_{\pi_i^*}\left\{\sum_{t=0}^{\infty}\gamma^t\left[r_t^e(s_t,a_t)+\beta r_{s_t}^i(s_t)\right]\right\} \\
\geq &\mathbf{E}_{\pi^*}\left[\sum_{t=0}^{\infty}\gamma^t r_t^e(s_t,a_t)\right]+\sum_{t=0}^{\infty}\gamma^t\beta\omega t.
\end{aligned}
\tag{17}
$$

From Eqns. (15) and (17), we can see that the equality sign in both equations hold. Since

$$
\mathbf{E}_{\pi_i^*}\left[\sum_{t=0}^{\infty}\gamma^t r_t^e(s_t,a_k)\right]=\mathbf{E}_{\pi^*}\left[\sum_{t=0}^{\infty}\gamma^t r_t^e(s_t,a_k)\right],
\tag{18}
$$

which proves that policy $\pi_i^*$ is at least a subset of policy $\pi^*$.

**Proof of existence for $\pi^*$ and $\pi_i^*$:** According to Bellman Equation Sutton & Barto (1998); Sutton et al. (1999), for any RL problem, the policy $\pi^*$ is exist. In addition, from Eqns. (16) and (18), we see that the policy $\pi^*$ can maximize the learning goal in Eqn. (4), so the policy $\pi_i^*$ is exist. $\qquad\square$

