# OpenReview forum: "DANCE: Difficulty and Novelty Co-driven Exploration"
_ICLR.cc/2026/Conference — ICLR 2026 Conference Withdrawn Submission_

### Official Review · Reviewer_ksRn · 2025-10-25

**Soundness:** 2
**Presentation:** 3
**Contribution:** 2
**Rating:** 4
**Confidence:** 4

**Summary:**

DANCE is an intrinsic-motivation method that blends novelty with enduring difficulty: a random-target vs. learned-predictor RPN yields intrinsic reward as prediction error, which is high for new states and, with revisits, converges to the state’s steps-to-reach (difficulty), thus sustaining exploration. A reward-shaping view argues policy invariance while explaining learning speed-ups; experiments show better maze coverage and stronger returns on hard-exploration Atari than RND/PPO.

**Strengths:**

- Recasts intrinsic motivation by combining novelty with a persistent, learned notion of state difficulty so that exploration remains guided even after novelty fades

- Presents a reward-shaping analysis showing that DANCE’s intrinsic reward is potential-based and thus policy-invariant to the original MDP optimum (despite being asymptotically inconsistent), strengthening correctness.

- The method is simple to implement

**Weaknesses:**

- Conceptual mismatch between “difficulty” and its proxy, the paper motivates difficulty as the hardness of reaching a state, yet uses normalized time step $d_s(s_t) \infty t$ as the proxy. This approximates hitting time only in special cases (fixed start, near-deterministic dynamics, minimal loops, consistent episode horizons). The paper does not characterize when this proxy fails (e.g., stochasticity, partial observability, multiple paths with different lengths, or environments with exogenous timers), beyond a brief “clock” limitation note

- The chosen proxy (step count) is heuristic; it may not always capture true task complexity or generalize well to all domains.

- The difficulty signal can be hijacked by irrelevant but predictable features (e.g., clocks in the environment), leading to misleading intrinsic rewards.

**Questions:**

- What is the relative contribution of novelty vs. difficulty? Would the method still be effective if one of the signals is removed?

- You argue the RPN error is high for novel states and, with revisits, converges to a state’s difficulty (steps-to-reach). Can you formalize conditions under which the prediction error e(s) decomposes as e(s)=novelty(s,t)+difficulty(s) and the difficulty component is identifiable (e.g., mixing/coverage assumptions on trajectories)?

- The reward-shaping analysis claims policy invariance despite DANCE’s intrinsic reward not vanishing asymptotically. Could you restate the exact potential $\phi(s_t)$ and clarify the scope of the invariance result: which choices of β (intrinsic weight), discount γ, and predictor training schedules still guarantee invariance? Any edge cases where off-policy updates or stale targets break the shaping argument?

- Table 1 compares DANCE to several exploration methods. Could you also include modern novelty/difficulty-style baselines (e.g., NovelD/NGU/Plan2Explore/E3B/DEIR where applicable) or clarify why they are out of scope? This would sharpen where DANCE’s “persistent difficulty” advantage matters most.

- In the shaping analysis and curriculum theorem, what assumptions on the behavior policy, stationarity, and predictor convergence are required? Can you provide a bounded-error statement: if the predictor tracks difficulty with error 𝜖, what’s the induced performance gap relative to the policy optimal under true difficulty?

---

### Official Review · Reviewer_duoK · 2025-10-31

**Soundness:** 2
**Presentation:** 3
**Contribution:** 2
**Rating:** 2
**Confidence:** 4

**Summary:**

This paper proposes DANCE (Difficulty and Novelty Co-driven Exploration), an intrinsic motivation mechanism for reinforcement learning in sparse-reward environments. The core idea is to combine state novelty and difficulty to maintain exploration signals throughout training. The method introduces a Reward Prediction Network (RPN), structurally similar to Random Network Distillation (RND), consisting of a fixed random target network and a predictor network trained to minimise prediction error. Unlike RND, the target embedding in DANCE is augmented with a “difficulty” signal proportional to the number of steps taken to reach each state. The resulting intrinsic reward is high for novel states early in training and converges to the state’s difficulty value as familiarity increases, thus preventing the intrinsic signal from vanishing. The authors provide theoretical analysis interpreting DANCE as a form of potential-based reward shaping that preserves policy invariance, and claim that the approach transforms sparse-reward problems into dense-reward ones. Empirical results on a maze domain and several hard-exploration Atari games suggest that DANCE improves exploration efficiency and learning stability compared to RND and other curiosity-based methods.

**Strengths:**

- The method is simple and understandable.
- The writeup is clean and correct.
- The visualisations show the exploration behaviour clearly.
- The algorithm seems to work reasonably well compared to other baselines.

**Weaknesses:**

* While a contribution doesn't have to be entirely novel to provide value, in my view DANCE is Random Network Distillation with an additional conditioning term attached to the target network based on the assumption that states further in time are harder to reach. This however I feel is a very simplistic assumption.
* The step index is not necessarily correlated with how hard a state is to reach. In many environments (e.g., stochastic, looping, or partially observable ones), a later timestep might correspond to:
	- wandering aimlessly in a known region,
	- revisiting trivial states, or getting stuck in local cycles.
* Also, difficult states may appear early (e.g., near traps, narrow passages, or high-risk transitions).
* One major issue I see is that by tying the difficulty to the time index rather than the structural properties of the state, DANCE seems to destroy the notion of reusability:
	- The same visual or semantic state encountered at different timesteps will yield different difficulty signals.
	- Consequently, the reward prediction network cannot learn reusable embeddings or abstractions.
	- This breaks compositionality and transferability, the model can’t generalise difficulty knowledge across trajectories or episodes.
	In contrast, a true difficulty measure (e.g., based on reachability, energy, or transition entropy) would be invariant to when the state was encountered.

* I believe the only truly difficult sparse exploration environment tested here was Montezuma's Revenge. I would have liked to see exploration on other sparse reward exploration domains such as MiniGrid or MiniHack, since they may uncover the shortcomings mentioned above (repeating states in partial observability).


Minor weaknesses:
* The abstract uses formulations that in my view make comprehension difficult. I would suggest the authors formulate the abstract in a more streamlined way.

**Questions:**

1. How exactly is the “difficulty” signal computed in practice, and does it adapt beyond being proportional to the timestep?
2. How would DANCE behave in partially observable or aliased environments (e.g., MiniGrid) where the same observation can appear at different timesteps with conflicting difficulty values?
3. Did the authors compare DANCE to a simpler baseline such as RND with a time-based or decaying intrinsic reward to isolate the effect of the difficulty term?
4. Can the claimed “curriculum formation” be demonstrated quantitatively rather than only through qualitative heatmaps?

---

### Official Review · Reviewer_6rN7 · 2025-11-01

**Soundness:** 3
**Presentation:** 2
**Contribution:** 2
**Rating:** 2
**Confidence:** 4

**Summary:**

The paper proposes DANCE, an intrinsic-motivation mechanism that combines novelty with a state-difficulty signal implemented via a Reward Prediction Network (RPN). The predictor is trained to match a target embedding and a scalar “difficulty” term tied to step count and similar to previous work such as RND, the intrinsic reward is the predictor–target discrepancy, which is high on novel states and converges to per-state “difficulty.” The authors also provided a theoretical analysis which frames the mechanism as potential-based reward shaping and argues policy equivalence with the original MDP.

**Strengths:**

1. Mixing novelty with a non-vanishing intrinsic signal to prevent late-stage drift is well motivated and easy to implement (RPN structure is close to RND).
2. This paper provides empirical results on hard exploration problems with an intuitive visualization. DANCE improves coverage in the pure-exploration Maze and shows competitive/strong results on several Atari benchmarks compared to baseline algorithms.

**Weaknesses:**

1. “Difficulty” lacks a formal, state-only definition. The paper sets difficulty proportional to the (normalized) step index t needed to reach a state and trains RPN so that the reward converges to that quantity. Acorss the paper does not present a formal and clean definition on the difficulty ds(s) beyond “proportional to steps.” in the text.
2.   Results are reported on one Maze and seven Atari games, but MiniGrid hard-suite} and ProcGen, which are natural tests for exploration and generalization are not included. The paper itself notes background-similarity issues (e.g., Solaris/Frostbite) that can degrade the step-learning signal, yet it does not demonstrate robustness beyond Atari.
3. Theorems 2 \& 3 are largely restate definitions (``the optimal policy maximizes the stated objective'') rather than offering new insight. If retained, they should be framed as preliminaries, not contributions. Convergence assumptions (for both the policy and the predictor) are not made explicit but are essential to the arguments.
4. Theorem 1 and Theorem 4 are not convincing:
        4.1): A ``fixed/converged'' policy may be only locally optimal and fail to visit many states; unvisited states invalidate the claim.
        4.2): Policy convergence does not imply the RPN has converged to its MSE optimum.
        4.3): Even if both converge, with a \(k\)-dimensional embedding and a scalar difficulty will broadcast to all dimensions,
        $r^{\text{int}}(s)=\|\hat g(s)-g(s)\|^2 = k\,\big[ds(s)\big]^2$
5. Theorem 4 may not hold as it depends on Theorem 1. It would benefit more from a clear existence proof.

**Questions:**

1. Can you compare NovelD on Atari and extend to MiniGrid/ProcGen to evaluate generality beyond Atari?
2. Is it guaranteed that the intrinsic reward will not dominate the training and mislead the policy learning?
3. Adding an explicit mathematical definition on difficulty measurement would be helpful for understanding the paper
4. Hyperparameter $\beta$ seems to be quite different in Maze and Atari game (from 0.01 to 0.5). How sensitive is your method to this hyperparameter?
5. It would be desirable to see the performance bound if the paper wants to focus on theoretical contributions.

---

### Official Review · Reviewer_R9fo · 2025-11-12

**Soundness:** 2
**Presentation:** 1
**Contribution:** 2
**Rating:** 2
**Confidence:** 3

**Summary:**

In this work, the authors consider the problem of intrinsic motivation in sparse-reward RL problems requiring substantial exploration. To complement state novelty (a signal which can lead agents to ignore “good” states if familiar, later in training), the authors propose to learn the “difficulty” of states. They show that if a difficulty metric can be defined, that policies learned based on this metric are a subset of optimal policies. They then show superior exploration in toy and several sparse reward Atari tasks.

**Strengths:**

* the idea of proposing a more stable metric for intrinsic exploration (other than novelty) is reasonable and commendable objective
* The authors do a decent job demonstrating that the metric they’re learning improves exploration on the tasks shown
* The authors provide several proofs which appear to be correct under the assumptions made.

**Weaknesses:**

* the paper is quite unclear in its description of the metric they are proposing
* the concept of “difficulty” is never clearly defined. And seems impossible to define universally, since certain policies will consider certain states “easy” and other states “hard”.
* Using “number of steps since start” or “number of steps away from start” requires a lot of assumptions on the environment which may not be there.
* not obvious how this would work with continuous control problems, where “difficulty” via distance is a lot murkier.
* In general, difficulty seems to require additional oracle information (dramatically limiting its utility), approaching providing something close to a dense reward / proper value function.

**Questions:**

* How does state novelty trade off with difficulty? No ablations were done here (e.g. tuning that tradeoff).

---

### Note · Authors · 2025-12-02

I have read and agree with the venue's withdrawal policy on behalf of myself and my co-authors.